# Co-Design and Validation of a Family Nursing Educational Intervention in Long-Term Cancer Survivorship Using Expert Judgement

**DOI:** 10.3390/ijerph20021571

**Published:** 2023-01-15

**Authors:** Marta Domingo-Osle, Virginia La Rosa-Salas, Ainhoa Ulibarri-Ochoa, Nuria Domenech-Climent, Leire Arbea Moreno, Cristina Garcia-Vivar

**Affiliations:** 1Faculty of Nursing, University of Navarra, 31009 Pamplona, Spain; 2Bioaraba, Osakidetza Basque Health Service, Araba University Hospital, 01009 Vitoria, Spain; 3Faculty of Health Sciences, University of Alicante, 03080 Alicante, Spain; 4Faculty of Medicine, University of Navarra, 31008 Pamplona, Spain; 5Faculty of Health Sciences, Public University of Navarre, 31008 Pamplona, Spain

**Keywords:** long-term cancer survivorship, family, interdisciplinary education, expert judgement, nursing education

## Abstract

The number of cancer survivors is increasing exponentially thanks to early screening, treatment, and cancer care. One of the main challenges for healthcare systems and professionals is the care of cancer survivors and their families, as they have specific needs that are often unmet. Nursing students, as future healthcare professionals, need education to face these new health demands. They will need to develop specific competencies to help them care for and empower this emerging population. The aim of the study was to co-design and validate an educational intervention on long-term cancer survivorship for nursing, through a multidisciplinary panel of experts. Group interviews were conducted with a panel of 11 experts, including eight professionals from different backgrounds (oncology, cancer nursing, pharmacology, and education), a long-term cancer survivor, a family member of a cancer survivor, and a nursing student. The experts validated a pioneer educational intervention to train nursing students in long-term cancer survival. The co-design and validation of the intervention from an interdisciplinary perspective and with the participation of long-term cancer survivors and their families was considered relevant as it included the vision of all the stakeholders involved in long-term cancer survivorship.

## 1. Introduction

There are currently 32.6 million cancer survivors worldwide [1]. In addition, there is an increasing number of long-term cancer survivors, i.e., individuals who are disease-free at 5 years after their diagnosis and the completion of their treatment, although many face late physical, psychological, and socioeconomic sequelae [2].

Therefore, the health system must accommodate the growing need for the long-term follow-up of survivors—who are considered chronic patients—to promote their well-being and improve their quality of life, to facilitate their return to work, and to enable them to live independently and to reduce their rate of cancer recurrence [3]. In order to improve the health outcomes and promote care that meets the needs of cancer survivors, health professionals should work in an interdisciplinary way for the comprehensive care of these survivors and their families [4]. Although cancer is a family concern [5], families do not receive the support they need and, consequently, they often experience the same distress that survivors experience [6,7]. Therefore, a comprehensive healthcare framework that addresses the family as the unit of care in cancer care is recommended [8].

The experience of managing cancer and the problems derived from it are complex, especially when the cancer patient and his or her family are at the center of care [9]. Hence, health professionals must be prepared to offer family-focused care in an interdisciplinary way [10]. In this research, the Calgary Model of Family Assessment and Intervention will be used as the theoretical framework as it is one of the most widely used and has been implemented in nursing education curricula worldwide [11].

New graduates will face health demands related to chronic care, such as that required by cancer survivors and their families. Thus, new graduates should develop specific skills to work as a team and to support, educate, and empower long-term cancer survivors and their families.

Studies have highlighted the need to train nursing students in the field of oncology [12]. They have also recommended the use of new educational methodologies in nursing, as well as the use of educational environments and tools such as clinical simulation to provide quality training in a safe context [13]. However, there is a lack of nursing curriculum models that address nursing practices for patients and families living with complex and long-term health processes, such as cancer [14]. In addition, no interdisciplinary educational intervention based on active methodologies to train nurses to provide family nursing care in the context of long-term cancer survivorship has been found. Some family interventions have been conducted in acute care areas such as the one conducted by Eggenberger et al. [15], where one of the central elements was therapeutic conversation between nurses and families in the intensive care unit, or the one developed by Beierwaltes et al. [16], which incorporated digital storytelling to implement family nursing practice in acute care settings. However, no educational intervention has been found in family nursing for learning to care for families in cancer survivorship.

Therefore, there is a need to develop educational interventions that enhance nursing students’ competencies in order to offer students opportunities to learn to assess and intervene in families who live with and beyond cancer, and to integrate this experience into their future clinical practices [17,18]. Furthermore, participatory design that involves people in the co-design of learning tools, educational policies, academic curricula, or support innovation processes has been recommended to ensure these works respond to the needs of the learners [19]. Therefore, the aim of this study was to co-design and validate a family nursing educational intervention in long-term cancer survivorship through an expert judgement.

## 2. Methods

### 2.1. Design

The expert panel method using qualitative focus groups with experts was used in this study to co-design and validate an interdisciplinary educational intervention. This was considered the most appropriate method because there were no useful previous data on which to base the educational intervention [20].

Unlike Delphi studies, whose objective is to reach a consensus on a given topic through the analysis of quantitative data, the expert panel method does not require a consensus to be reached [20]. Its objective is to obtain opinions from specialist experts and other individuals of interest, and the information that emerges from the group dialogue and interaction is used to validate a product, in this case, the methodological and content design of an educational intervention. The panel evaluation of teaching material consisted of asking a group of experts to discuss the advantages and disadvantages of an educational program (learning objectives, methodology, content, and evaluation) and reach a consensus on the best features of an educational program. This evaluation strategy provides a deeper assessment and detailed information on the subject under study [21].

### 2.2. Data Collection

#### 2.2.1. Expert Selection

Regarding the procedures for selecting the experts, diversity of opinions are found in the literature, ranging from those that do not imply any selection filter—as in the cases of an affinity or closeness between the expert and the researcher—to those that use a series of structured criteria such as the Biogram or the Expert Competence Coefficient [22]. The present study was carried out by affinity with the research group (belonging to different institutions with which the authors had worked previously).

As for the number of experts needed to make up the panel, there is no unanimous agreement on its determination [22]. Some authors point out that the number of experts depends on aspects such as the ease of accessing them or the possibility of knowing enough experts on the subject under investigation [21]. Other authors indicate that the number of experts in a panel depends on the level of expertise and the diversity of knowledge [23]. In this study, specialization was taken into account (all had to be working in the field of oncology either in the clinic or university) and knowledge should include both the main health professionals involved in the care of cancer survivors and the survivors and family members themselves. This is in addition to university lecturers specializing in educational methodologies and the students who will receive the intervention.

A convenience sample was used to select the participants based on the representation of the different roles and profiles necessary for an educational intervention based on the expertise of the main actors. The panel was comprised of 11 participants from different institutions; of which, 8 were specialist experts (5 healthcare practitioners and 3 academic professors) with scientific knowledge and care experience on the subject under study (long-term cancer survivorship care), 2 were patients/family members, and 1 was a potential intervention recipient (a nursing student at the doorstep of graduation) Table 1.

The expert panel of participants were recruited in Spain from the areas of primary care, hospitalization, research, and education (Table 2).

#### 2.2.2. Conducting the Expert Panel

The ways of developing expert panels are diverse; in this study, the consensus method was used. As a group and jointly, the participants reached an agreement [21]. In this study, it was necessary for 80% of the panelists to agree. The panel was established in December 2020 and the meetings were held on the 15th and the 22nd of January 2021. Each meeting had a duration of one hour.

The expert panel was structured in three consecutive stages. The first stage involved constitution and information, and the second and third involved group meetings for a discussion and drawing conclusions (Figure 1).

#### 2.2.3. Stage 1. Constitution of the Expert Panel, Final Expert Selection, and Information

Experts were invited to participate by telephone and email. Following recommendations by Lecours et al. [24], the email message contained the following information: principal investigator and research team, description of the study, reasons why the expert was selected, procedure to follow to participate in the panel, estimation of the time required (participation in all stages), and confidentiality. All of the experts who were contacted voluntarily agreed to participate in the study, signed the informed consent form, and confirmed their attendance before the first meeting. In response to their acceptance, the experts were sent a file explaining the proposal for the interdisciplinary educational intervention, as well as the link to the videoconference for the first meeting. The file contained the explanation and justification of the study, the composition and characteristics of the expert panel members, and the proposed educational intervention to be validated.

#### 2.2.4. Stage 2. First Expert Panel Meeting

The group moderator (principal investigator) thanked all members for their participation and for dedicating their time to the study. She then introduced all of the members and made a brief presentation of the intervention proposal submitted for validation, including the objectives, contents, and methodologies. Next, the experts were asked to provide their inputs regarding the need for the project and its objectives, content, and methodologies, and the discussion began. The first meeting concluded with a brief review of the topics discussed. The moderator also confirmed that minutes of the meeting would be sent out, the date and time of the next meeting was set, and some questions for reflection were provided, focused on helping panelists draw conclusions about the educational intervention.

#### 2.2.5. Stage 3. Second Expert Panel Meeting: Content Validation of the Intervention, Experts’ Opinion, and Proposed Changes

The principal investigator read the minutes of the previous meeting, as well as the questions that were sent to the panel members for individual work. Some of them were: what would you like nursing students to know about cancer survivorship and the family care of cancer survivors?; do you consider that the proposed educational intervention helps to acquire competence for cancer survivor and family-focused care?; what kind of content do you suggest for the personal work of the students?; what educational methodologies would you use in the training of students?; what do you think is currently missing in the training of nurses for the care of cancer survivors and their families?; what contents do you suggest for students’ personal work?; and what else do you think nursing professionals need to know?. All the members of the panel gave their inputs. However, because the cancer survivor could not attend the second meeting due to personal reasons, she sent in writing her contributions to share them with the group. After discussing and analyzing all the contributions, a consensus was reached regarding the design of the interdisciplinary educational intervention, which was called “Learning and Care”.

### 2.3. Data Analysis

A thematic analysis was carried out to identify meaningful pieces of qualitative data from the transcriptions of the two meetings with the experts [25]. Thematic analysis is used to explore experiences, perspectives, and practices and to analyze the qualitative data collected from interviews, focus groups, or surveys, among others [25]. In this study, thematic analysis was used to analyze the perspectives and opinions of experts regarding the design and content of an educational intervention for nursing students. The six steps outlined by Braun and Clarke [25] were as follows: (1) one of the authors (MD) read the full data material closely to become familiar with what the data entails, paying specific attention to patterns that occur; (2) through a data reduction, MD started to generate the initial codes by documenting where and how patterns occur; (3) at this stage, two of the researchers (MD and VL) met for an analytic seminar to combine codes into overarching themes; (4) at this stage, the two researchers discussed a coherent recognition of how the themes were patterned; (5) MD and VL defined each theme which emerged and captured how the themes supported the data; and (6) finally, CGV, who had not been part of the analysis process thus far, audited the final thematic structure and its relationship with the identified data from the participants. (Table 3 analysis process).

### 2.4. Rigor

The rigor of this study was ensured following recommendations by Doyle et al. [26]. The credibility was ensured following a data analysis developed by three qualitative experts in the research team and was supported by the quotes of the panel experts. Furthermore, the data analysis was guided by reflexivity where previous assumptions were recognized and left in suspension. The reliability was guaranteed presenting a detailed description of the methods. Transferability and confirmability were safeguarded by presenting detailed information about the participants sociodemographic data and the research scenario.

### 2.5. Ethics Committee Approval

The present study, which is part of a larger study, received ethical approval (ref. 2020.161), thus ensuring that it respects the fundamental principles of the Declaration of Helsinki. All participants were aware of the study, voluntary participated, and signed the corresponding informed consent form. In addition, anonymity, confidentiality, and results communication were guaranteed throughout the research.

## 3. Results

The results are structured according to the three main themes that emerged from the collected data.

### 3.1. Experts’ Opinions Regarding the Content in the Educational Intervention

To acquire competence (knowledge, skills, and attitudes) in family-focused care for long-term cancer survivorship, the following learning objectives were proposed:Understand the needs of long-term cancer survivors and their families.Know the characteristics of the family interview according to the Calgary Family Assessment and Intervention Model [27].Acquire the ability to conduct a family interview according to the Calgary Model.Encourage an attitude of care focused on the cancer survivor and his or her family.Encourage interdisciplinary work that promotes family-focused care in cancer survivorship.

Once the objectives were presented and agreed upon, the contents to be taught during the educational intervention were discussed. Additionally, different educational methods for covering all of the competency dimensions were identified.

One of the proposed contents was related to the interdisciplinary work, as stated by the oncologist: “*I think it is important to address interdisciplinary work in a round table and how it affects the care of survivors and their families….At the end of the round table, before leaving the classroom, students should be able to answer some questions with their phone or electronic device to encourage them to reflect on what they have heard, mainly about interdisciplinary work.”*

On the other hand, the psycho-oncologist indicated the importance of including new content so that the students would be aware of the fear of recurrence that survivors and relatives have and stated: “*Patients and family members are afraid of recurrence. Therefore, this concept (fear of recurrence of cancer) must be present in the intervention and the students must learn to give patients and relatives realistic hope.”*

Subsequently, to understand the importance of interdisciplinarity and to delve deeper into the needs and experiences of cancer survivors and their families, the need for an exchange of experiences among survivors, family members, health professionals, and students was agreed upon.

Finally, to integrate the acquisition of the full competence, the experts proposed the inclusion of a clinical simulation in which the learners could experience a “real” assessment and care situation with cancer survivors and their family.

### 3.2. Combination of Innovative Teaching Methods

The expert panel considered it appropriate and relevant to use a combination of innovative methods in the educational intervention to deliver the content appropriately, including the following three methods: a flipped classroom, round table, and clinical simulation.

The flipped classroom—which was recommended by the oncologist for the acquisition of knowledge—*enables learners to be leaders of their learning to facilitate their clinical reasoning and their critical thinking skills*. The expert panel suggested that the contents should be dynamic, clear, and brief, and should be delivered to the learners via videos, TEDx conferences, and research articles. It was agreed that the flipped classroom would be taught by a clinical nurse with knowledge and skills in conducting family interviews in nursing practice. Additionally, it was suggested by the nurse practitioner that the class should use didactic tools, such as Kahoot, role playing, and group dynamics, among others.

The round table format—which was recommended to address the nurses’ attitudes—seeks to emphasize the interdisciplinary work and how it affects the care of survivors and their families. The round table would be composed of an oncologist, an oncology nurse, a long-term cancer survivor, and a family member of the survivors who would narrate their experiences. Furthermore, as stated by the nurse researcher “*It is good to allocate a long time to the round table to facilitate questions from the students to the speakers*”. This statement, which was agreed upon by all the members of the expert panel, was incorporated into the educational intervention. It was also suggested to send to the cell phones of students some questions to promote their personal reflection on the topics addressed in the round table.

The clinical simulation—which was recommended for the development of skills—would consist of a family interview and a therapeutic conversation with a long-term cancer survivor and his/her family member. This clinical simulation would bring students and new graduates closer to contexts similar to those that they will encounter in practice and allow them to apply their knowledge, acquire desired attitudes, and develop their skills. In addition, as stated by the student: “*Simulation is something we students really enjoy because it gives us the opportunity to practice before going to clinical practice”*. Even though all the panelists agreed on the importance of clinical simulation, it was the most controversial methodology among the experts. Debate was opened on whether the clinical scenario should be carried out with real or standardized patients/relatives and whether it should take place in person or via a videoconference (due to the COVID-19 pandemic). It was ultimately considered appropriate to use standardized patients and promote the involvement of the student in a clinical simulation to build interpersonal and face-to-face relationships between students and survivors/family members.

The combination of different educational methods and the content of each one was well accepted by the expert panel, reaching a consensus of more than 80% of the participants. According to the psycho-oncologist who was supported by all experts, “*the educational intervention is very well designed, very complete and contributes to the acquisition of skills to care for cancer survivors and their families*”. The student also stated: “*I am interested in having different activities and methods because I believe that each one brings something different to the table*”. Finally, the survivor and the family member highlighted that “*the direct involvement of survivors and family members in nursing education is a positive factor”.*

The educational intervention validated by the expert panel is outlined in Figure 2.

### 3.3. Need of Education in Long-Term Cancer Survivorship

During the first meeting, the expert panel unanimously expressed the need to train students in the care of cancer survivors and their families. The survivor considered this *“very positive for the quality of life of cancer survivors, that nursing is present in the stage of cancer survivorship as much as it is during treatment”.* The survivor reaffirmed that she considered it important not only for her and her family but also for *“the economic repercussion that this has for society due to the frequency of sick leave or even partial disability that this situation may result in, which perhaps, with nursing care, could be reduced”.*

Similarly, the oncologist, psycho-oncologist, pharmacist, and nurses expressed “*the need to train nurses to be able to carry out a long-term follow-up similar to that performed during the active phase of cancer treatment*”. The family member agreed on the importance of education in the area of cancer survivorship, although she argued that “*No one has cared about me as a family member*”.

Finally, the four-year nursing student who was close to graduating considered that “*I still need to know more about cancer survivorship and family-focused care. It is discussed in class (in theory), but I have not had the opportunity to learn about it in practice (in a clinical simulation)*”.

Thus, all the experts agreed on the importance of nursing education in long-term cancer survivorship with a family-focused approach.

## 4. Discussion

This study presents the characteristics and benefits provided by the collaboration of an interdisciplinary panel of experts to co-design and validate an educational intervention on long-term cancer survivorship for nursing. The expert judgement method has been widely used to validate research tools and educational interventions that need to be carried out rigorously and have not been done before [28]. It is considered to be of importance to establish the validity of the content in educational interventions in the area of health, since they will influence the quality of the learning outcomes [29].

Nursing education on long-term cancer survivorship is emerging, so it was considered appropriate to validate a new interdisciplinary educational intervention before implementing it. Involving patients and family members in education brings benefits such as bridging the gap between theory and clinical practice [30].

Regarding this study, a consensus was used to develop the panel, which was reached without problems (always between 50% and 80%). This consensus may be due to the variety of panelists, which guaranteed complementary points of view [23].

In addition, all experts highlighted the importance and need for nursing education in cancer survivorship. This need was also identified by the Institute of Medicine (2006) in the report “*From cancer patient to cancer survivor: Lost in transition”*, which noted that cancer survivorship care should be included in the content of continuing health education, including education for nurses, physicians, rehabilitation specialists, and psychosocial and mental health professionals. Along these lines, Klemp et al. [31] stated that the majority of undergraduate nursing students receive little or no education to meet the needs of cancer survivors. Dietmann [32] also noted that although the number of people who survive cancer continues to increase, and the short- and long-term effects of cancer and its treatment result in physical, psychosocial, and spiritual needs, this content has not been addressed in nursing curricula. Therefore, we consider it essential to conduct future studies that demonstrate the effectiveness of nursing education to improve care for long-term cancer survivors.

The panel of experts provided creative multidisciplinary perspectives and allowed for the credibility, future acceptability, and application of the educational intervention, as stated by Dinessen et al. [33].

The panelists considered it necessary for the intervention to be taught interdisciplinarily and with different educational methods and dynamic content. To provide quality care in oncology, professionals must work collaboratively and make joint decisions [34]. Additionally, interdisciplinary education helps to increase the quality of care by improving nurses’ attitudes and perceptions regarding other healthcare professions [35]. Using complementary active educational methods was also found to be beneficial to acquire the different dimensions of competency (knowledge, skills, and attitude) to provide comprehensive care for long-term cancer survivors and their families. This was also highlighted by Dietmann [32] who indicated the importance of implementing active teaching methods for cancer education in undergraduate and graduate nursing curricula.

This study has strengths and limitations. The heterogeneous composition of the expert panel stands out as a strength: it encompassed experts from various disciplines who work together to provide comprehensive cancer care. Furthermore, including a cancer survivor and a family member (recipients of care) and a senior nursing student was positive to develop a more realistic educational intervention to be implemented.

However, it should be noted that the educational intervention may be brief to ensure the achievement of the full competence to care for long-term cancer survivors. More education is required to develop sufficient skills and knowledge to address the specific needs experienced by long-term cancer survivors and their families.

## 5. Conclusions

This study describes the process of an expert panel to co-design an educational intervention in the long-term cancer survivorship for nurses. The educational intervention will be interdisciplinary and will use three different active educational methodologies: a flipped classroom, roundtable, and clinical simulation. It will bring students closer to the needs of long-term cancer survivors and their families, help them become aware of their own learning needs, and train them in the assessment and care of these individuals through family interviewing and therapeutic conversation. This work emanates from clinical practice, the unmet needs of cancer survivors and their families, and the need for teamwork in oncology. Finally, it will help to advance the education of future nursing professionals and therefore improve their clinical practice.

## Figures and Tables

**Figure 1 ijerph-20-01571-f001:**
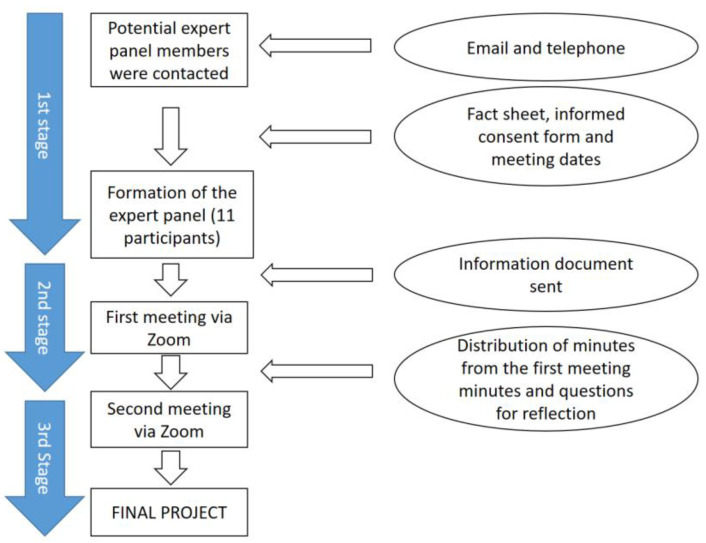
Flow chart of the expert panel.

**Figure 2 ijerph-20-01571-f002:**
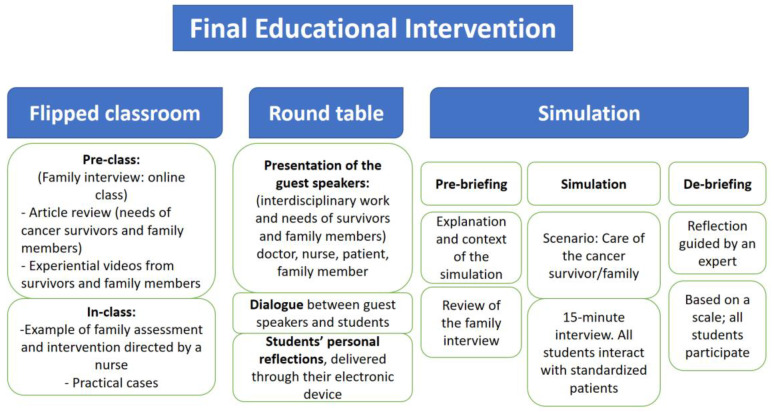
Final Educational Intervention.

**Table 1 ijerph-20-01571-t001:** Selection Criteria.

Inclusion Criteria	Exclusion Criteria
Specialists in the area of oncology.	Professionals with less than 10 years of experience (because of their short professional trajectory in this area).
Currently working with cancer survivors and their families.	Relatives and survivors with less than 5 years free of disease (as they are not considered to be long-term cancer survivors).
Teaching experience in the area of health sciences.	Early career nursing students.
Cancer survivors who have been untreated for at least 5 years.	
Family members of cancer survivors who have been untreated for at least 5 years.	
Cancer survivors and families with sufficient academic training to actively participate in the project and in the expert panel.	
Being able to complete the two meetings via Zoom.	
Senior nursing students.	

**Table 2 ijerph-20-01571-t002:** Expert Panel Members.

Expert	Areas of Knowledge	Main Role	Years of Professional Experience/Years of Survivorship
1.	Nursing	Primary care nurse *	Over 20 years
2.	Nursing	Hospital nurse **	Over 10 years
3.	Nursing	University professor of nursing **	Over 10 years
4.	Nursing	University professor of nursing ***	Over 15 years
5.	Nursing	University professor of nursing **	Over 20 years
6.	Medicine	Radiation oncologist ****	Over 15 years
7.	Pharmacy	Oncology pharmacist	Over 10 years
8.	Psychology	Psycho-oncologist	Over 10 years
9.	Patient	Breast cancer survivor	10 years
10.	Family member	Daughter of a colon cancer survivor	15 years
11.	Senior student	Four-year university nursing student	4 years

In addition to the described role, the individual also exercises the following role: * University professor. ** Researcher in cancer survivorship. *** Researcher in innovative teaching methods. **** Expert in innovative teaching methods.

**Table 3 ijerph-20-01571-t003:** Analysis process.

Declaration	Category	Theme
*“It is key to make good use of the simulation to learn how to do a family interview, for this the previous readings are important. Provide students with articles about the 15-minute Calgary Model interview”* (Nurse researcher).	Training contents	Opinions of the experts on the content of the educational intervention
*“The contents have to be dynamic, clear and brief and that they are delivered to the students through videos, TEDx conferences and research articles”* (Nurse teacher).		
*“It is necessary to know the needs and experiences of cancer survivors and their families”* (All experts).		
*“It seems important to me to address interdisciplinary work in a round table and how it affects the care of survivors and their families”* (Oncologist).	Need for interdisciplinary work	
*“At the end of the round table, before leaving the classroom, students can answer some questions with their phone or electronic device to encourage them to reflect on what they have heard, mainly about interdisciplinary work”* (Oncologist).		
*“Patients and family members are afraid of recurrence. For this reason, it is necessary that this concept is present in the intervention and the students learn to give patients and family members a realistic hope”* (Psycho-oncologist).	Knowledge of fear of recurrence	
*“The educational intervention is very well designed, it is very completeand contributes to the acquisition of skills to care for cancer survivors and their families”* (Psycho-oncologist).	Good design and varied to acquire competence	Combination of innovative teaching methods
*“I really liked the intervention. I think it covers everything we need to acquire full competence. I am interested in having different activities and methods because I believe that each one contributes something different”* (Student).		
*“I recommend the flipped classroom, for the acquisition of knowledge, it allows students to be leaders of their learning, it facilitates clinical reasoning and critical thinking” *(Oncologist). *“In addition, I propose that the flipped classroom be taught by a clinical nurse with knowledge of the family interview”* (Primary care nurse).	Student-led educational methods	
*“Simulation is something that we students like a lot because it gives us the opportunity to practice before going to the clinical practice”* (Student).		
*“It is good to allocate a long time to the round table to facilitate the students’ questions to the presenters”* (Nurse research).	Dialogue to foster learning	
*“We consider that the direct participation of survivors and family members in nursing training is a positive factor”* (Survivor and family).		
*“Training is very necessary, it is very positive for the quality of life of cancer survivors. That nursing is present both in the cancer survivor stage and during treatment”* (Cancer survivor).	Need for training to improve quality of life	Need for training in long cancer survival
*“I consider training important, not only for the survivor and their family, but also because of the economic repercussions that this has for society due to the frequency of sick leave or even partial disability that this situation can cause, which perhaps, with emergency care nursing, could be reduced”* (Survivor).		
*“It is necessary to train nursing professionals in the care of cancer survivors and their families to be able to carry out a follow-up similar to that carried out during the active phase of cancer treatment”* (Oncologist,).	Need for training to improve accompaniment	
*“Now… where is the nurse who takes care of my dad? Training is needed in this field. I clearly see the need for training in the area of cancer survivorship. No one has cared about me as a relative”* (Family member of cancer survivor).		
*“In my undergraduate training, I still need to know more about cancer survivorship and family-focused care. It is discussed in class, but I have not had the opportunity to learn it in practice” (in a clinical simulation)* (Student).	Need for training in the absence of undergraduate studies	
*“There is a need for trained nurses who open doors to survivors and their families and introduce them to the system, help them navigate the process, and anticipate the needs of both survivors and family members. The existence of a gap in training in cancer survivorship and family nursing has been well established”* (Nurse professor).		

## Data Availability

The data presented in this study are available on request from the corresponding autor.

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
