# Peer review of "Co-Design and Validation of a Family Nursing Educational Intervention in Long-Term Cancer Survivorship Using Expert Judgement"

_ijerph, 2023, doi:10.3390/ijerph20021571_

Round 1

Reviewer 1 Report

Dear Authors,

I would like to congratulate you for the work described in the current manuscript. Although I found it interesting and of value, there are a few methodological aspects that I believe should be explained and ammended.

My recommendations in the pdf file aim to improve those aspects.

Here is my main concern. Although the work claims to be a content validity assessement of an intervention, the methodology is not followed congruently to allow for such an endeavour. I beleive this work is worth publishing as it entails the co-design of an interventions with experts. However, the writing in its current state diminishes the value of the work, it is not a matter of language typos, is the argumentation and connection between arguments. Moreover, the methdology should focus on the expert panel to co-design of interventions, rather than on the content validity assessement, and the title needs to be re-written to eliminate the content validity component.

Also, the authors should also consider to bear in mind gold standards for conducting and reporting research, for example the SRQR (cf. Equator Network).

Reviewer 2 Report

Grateful for the opportunity to review this manuscript.

This study is relevant to the teaching area and can be translated into better-prepared students in the area under investigation.

I only offer small suggestions:

- Throughout the article, citation rules vary between those requested by the journal and APA (see lines 66, 123, 278,…). The referencing standard must be standardized.

- This is a qualitative study; however, the authors do not refer to the rigour and quality criteria used. This information would enrich the study from a methodological point of view.

- I suggest mentioning the legal and ethical aspects complied with (this is, in my opinion, an essential element missing from the article). How was the anonymity of participants ensured?

- The Conclusion section lacks a clear answer to the research question/objective.

Best wishes for a good job.

Reviewer 3 Report

It is very important to develop nursing education contents and methods for cancer survivors.

In addition, I salute the researchers' spirit of challenge to secure the validity of the contents and methods of education through expert panel research.

1. However, since qualitative research methodology was used, the following information should be included.

In particular, for Rigor of the expert panel study, please suggest efforts to remove the panel biases in the research method.  For examples of panel bias:

1) A wide enough selection of experts to cover all aspects of scientific thinking on the subject

2) Mindset – Unspecified Assumptions Used by Experts

3) Structural bias – at level of detail or background scale selection for quantification

4) Motivational bias – Experts have a stake in research findings.

5) cognitive bias;

 Overconfidence – manifests in uncertainty estimation.

 Anchorage – Experts subconsciously make judgments based on previously given estimates.

Availability – where memorable (difficult) events are likely to be overestimated (underestimated).

2. Include the following in your research methods.

: Please describe the main questions in more detail during the panel meeting.

3. Also, describe the qualitative data analysis method (coding method and topic derivation method when analyzing content) with more detailed examples.

Summarize the following in your findings. (Please be more specific in the supplementary)

: Statement, coding, theme (Table or Summary of contents)
